# The Oral–Gut Microbiota Axis Across the Lifespan: New Insights on a Forgotten Interaction

**DOI:** 10.3390/nu17152538

**Published:** 2025-08-01

**Authors:** Domenico Azzolino, Margherita Carnevale-Schianca, Luigi Santacroce, Marica Colella, Alessia Felicetti, Leonardo Terranova, Roberto Carlos Castrejón-Pérez, Franklin Garcia-Godoy, Tiziano Lucchi, Pier Carmine Passarelli

**Affiliations:** 1Geriatric Unit, Fondazione IRCCS Ca’ Granda Ospedale Maggiore Policlinico di Milano, 20122 Milan, Italy; do.azzolino@hotmail.it (D.A.); tiziano.lucchi@policlinico.mi.it (T.L.); 2Respiratory Unit, Cystic Fibrosis Adult Center, Fondazione IRCCS Ca’ Granda Ospedale Maggiore Policlinico di Milano, 20122 Milan, Italy; margherita.carnevale@policlinico.mi.it (M.C.-S.); leonardo.terranova@policlinico.mi.it (L.T.); 3Interdisciplinary Department of Medicine, Section of Microbiology and Virology, School of Medicine, University Hospital of Bari, 70124 Bari, Italy; luigi.santacroce@uniba.it (L.S.); marycolella98@gmail.com (M.C.); 4Doctoral School, eCampus University, 22060 Novedrate, Italy; 5Department of Medical and Surgical Sciences, Magna Graecia University, 88100 Catanzaro, Italy; felicettialessia21@gmail.com (A.F.); 6Instituto Nacional de Geriatría, National Institutes of Health, Ministry of Health, Ciudad de México 14080, Mexico; rc.castrejon.perez@gmail.com (R.C.C.-P.); 7Bioscience Research Center, College of Dentistry, University of Tennessee Health Science Center, Memphis, TN 38163, USA; fgarciagodoy@gmail.com (F.G.-G.); 8The Forsyth Institute, Cambridge, MA 02142, USA; 9Department of Surgery, Herbert Wertheim College of Medicine, Florida International University, Miami, FL 33199, USA; 10Department of Head and Neck and Sensory Organs, Division of Oral Surgery and Implantology, Fondazione Policlinico Universitario A. Gemelli IRCCS, University Cattolica del Sacro Cuore, 00168 Rome, Italy; piercarmine.passarelli@unicatt.it (P.C.P.)

**Keywords:** microbiome, oral microbiota, gut microbiota, gut dysbiosis, systemic diseases, inflammation, diet, probiotics, prebiotics

## Abstract

The oral–gut microbiota axis is a relatively new field of research. Although most studies have focused separately on the oral and gut microbiota, emerging evidence has highlighted that the two microbiota are interconnected and may influence each other through various mechanisms shaping systemic health. The aim of this review is therefore to provide an overview of the interactions between oral and gut microbiota, and the influence of diet and related metabolites on this axis. Pathogenic oral bacteria, such as *Porphyromonas gingivalis* and *Fusobacterium nucleatum*, can migrate to the gut through the enteral route, particularly in individuals with weakened gastrointestinal defenses or conditions like gastroesophageal reflux disease, contributing to disorders like inflammatory bowel disease and colorectal cancer. Bile acids, altered by gut microbes, also play a significant role in modulating these microbiota interactions and inflammatory responses. Oral bacteria can also spread via the bloodstream, promoting systemic inflammation and worsening some conditions like cardiovascular disease. Translocation of microorganisms can also take place from the gut to the oral cavity through fecal–oral transmission, especially within poor sanitary conditions. Some metabolites including short-chain fatty acids, trimethylamine *N*-oxide, indole and its derivatives, bile acids, and lipopolysaccharides produced by both oral and gut microbes seem to play central roles in mediating oral–gut interactions. The complex interplay between oral and gut microbiota underscores their crucial role in maintaining systemic health and highlights the potential consequences of dysbiosis at both the oral and gastrointestinal level. Some dietary patterns and nutritional compounds including probiotics and prebiotics seem to exert beneficial effects both on oral and gut microbiota eubiosis. A better understanding of these microbial interactions could therefore pave the way for the prevention and management of systemic conditions, improving overall health outcomes.

## 1. Introduction

The oral microbiota has generally been less explored than the gut microbiota. However, in recent years, there has been a growing interest in research on the oral microbiome [1], becoming one out of five research priorities of the human microbiome project [2]. Research on the gut microbiota has benefited from advanced technologies and increased availability of fecal samples, which have made studies easier and more direct [3]. However, the oral microbiota is equally important in determining both oral and systemic health since it represents the first entry door for nutrients and pathogens in the digestive system [1,4].

The exploration of the oral microbiota is recently growing, thanks in part to the use of advanced sequencing technologies and increasing awareness of the importance of maintaining a balanced microbiota [1]. Research is beginning to elucidate how dietary habits, oral hygiene, and environmental factors can influence the composition and function of the oral microbiota, and how these changes may be linked to oral and systemic diseases [5,6]. In this context, diet plays a pivotal role in shaping both the oral and gut microbiota composition. The variety of dental problems associated with oral dysbiosis alter food choices leading to the consumption of soft and easy-to-chew foods, frequently rich in saturated fats and refined sugars but lacking essential beneficial nutrients (e.g., proteins, fiber, vitamins, and minerals) [7,8,9] with the consequent disruption of gut microbiota homeostasis [10]. Some nutrients, bioactive compounds and dietary patterns as well as certain probiotic and prebiotic strains seem to have beneficial effects on both the oral and gut microbiota. Furthermore, key microbial byproducts and derived bioactive components, such as short-chain fatty acids (SCFAs), trimethylamine *N*-oxide (TMAO), indole and its derivatives, bile acids, and lipopolysaccharides (LPSs), seem to play central roles in mediating various pathological conditions, by acting on inflammation, oxidative stress, lipid and glucose metabolism, and body barrier integrity. There is growing research interest in the interaction between the oral and the gut microbiota within three main routes of communication namely the enteral, the hematogenous and fecal–oral routes [11]. The aim of this review is therefore to provide an overview of the interactions between oral and gut microbiota, and the influence of diet and related metabolites on this axis.

## 2. Overview of the Oral and Gut Microbiota Composition Across the Lifespan

The oral microbiota is a complex and dynamic ecosystem composed of a wide range of microorganisms, including bacteria, viruses, fungi, and archaea [12]. The Human Oral Microbiome Database (eHOMD) is one of the most comprehensive databases on oral human microbiota [13]. Figure 1 shows the main relative % phyla of oral microbiota stored in the HOMD.

Bacteria are the predominant components of the oral microbiota and represent a wide variety of species (i.e., over 700) [12]. The oral cavity is composed of both oxygen-deprived areas (e.g., subgingival surfaces) and relatively oxygen-rich areas (e.g., supragingival surfaces), thus supporting both aerobic and anaerobic bacterial species [14]. The main bacterial phyla in the oral microbiota include *Bacillota*, *Actinomycetota*, *Pseudomonadota*, *Bacteroidota*, and *Spirochaetota*, with *Spirochaetes* being the main genus [15]. The phylum *Bacillota* includes bacteria such as *Streptococcus*, which are among the most abundant in the oral cavity and are involved in dental plaque and caries formation [16]. Other members include *Lactobacillus*, which may also be implicated in dental caries formation [17]. *Actinomycetota* such as *Actinomyces* spp. and *Bifidobacterium* spp. play important roles in the balance of oral microbiota [18]. *Pseudomonadota* including *Neisseria* are commonly found in the oral mucosa and saliva [19]. Some bacteria in the *Bacteroidota* group, such as *Prevotella* spp., are associated with gum disease and periodontal disease [20]. In *Spirochaetota*, spirochetes such as *Treponema* are related to advanced periodontal disease and other oral infections [21]. Fungi, especially those belonging to the *Candida* genus (such as *Candida albicans*), are present in small amounts in the oral microbiota. However, *Candida albicans* can become pathogenic, causing infections such as oral candidiasis, especially in immunocompromised individuals [22].

The oral microbiota also hosts a series of viruses, including phages (viruses that infect bacteria). Phages can influence the composition of the oral microbiota by preferring or eventually killing specific bacteria [23]. Additionally, viruses such as the Human Papillomavirus (HPV) may also be present in the oral cavity and have been associated with oral cancer [24]. Although present in smaller amounts than bacteria, *Archaea*, such as those in the genus *Methanobrevibacter*, are part of the oral microbiota [25,26]. Some studies suggest that they may play a role in dental plaque formation, periodontitis and bad breath (i.e., halitosis), as some genera of *Archaea* are implicated in methane production [26,27,28].

It has been reported that oral microbiota composition, once established, remains relatively stable over time [29]. However, several factors across the life course including diet, oral hygiene, and hormonal changes can lead to oral dysbiosis [5]. At birth, the composition of the oral microbiota is mainly influenced by gestational age (i.e., full-term vs. premature), type of delivery (i.e., cesarean section vs. vaginal delivery), whether the infant is breastfed or formula-fed, maternal nutritional status, and exposure to antibiotics [30]. During this period, there is a shift from a relatively simple microbiota to a more complex one as the child grows, begins tooth eruption and starts eating solid food. Indeed, it has been suggested that the establishment of the adult oral microbiota seems to be determined by 18 months of age [31]. Hormonal changes as well as oral hygiene during puberty can also impact the oral microbiota [32]. During the adolescence period, oral bacteria can proliferate, particularly with the rise in saliva production and changes in the pH of the mouth. Teenagers may also have higher levels of gingivitis or dental plaque due to inconsistent oral hygiene practices [33]. In adulthood, the oral microbiota becomes more stable and diverse, influenced by factors like diet, oral hygiene habits, lifestyle choices (e.g., smoking, alcohol consumption), medication use (particularly antibiotics), or underlying pathological conditions (e.g., diabetes, gum disease) [19].

With aging, the diversity of the oral microbiota may decrease [34]. In particular, the number of beneficial bacteria may decline, while the abundance of potentially harmful bacteria can increase, contributing to oral diseases like gum disease, dry mouth, and tooth decay [34]. Oral commensal *Neisseria* tends to decline with aging (i.e., after the age of 40), while lactobacilli, *Streptococcus anginosus* and *Gemella sanguinis* species increase after the age of 60 [35]. Older adults are also more likely to have dentures or other dental appliances, which can further alter the oral microbial community [34]. In fact, notable differences have been found in both the composition and diversity of the oral microbiota, especially between older adults wearing dentures versus those having natural teeth. In denture wearers, bacilli and actinobacteria have been predominantly found at denture and oral mucosa levels [36]. Furthermore, having residual natural teeth significantly impacts oral microbiota composition and diversity of denture wearers [36]. Older people with less natural teeth show predominant commensals such as *Prevotella histicola*, *Veillonella atypica*, *Streptococcus salivarius* and *Streptococcus parasanguinis* [34,37]. However, it has been reported that toothy centenarians (i.e., those with 20 or more remaining natural teeth, also called ‘successful oral agers’) have a more diverse oral microbiota [38]. In particular, the oral microbiota composition of toothy centenarians is characterized by a predominance of the phyla *Spirochaetota* and *Synergistota*, of the genus *Aggregatibacter*, *Prevotella* spp., *Campylobacter* spp., *Anaeroglobus* spp., *Selenomonas* spp., and *Fusobacterium* spp., and of the *Porphyromonas endodontalis* species, found both in plaque and saliva [38]. A higher relative abundance of genera like *Bifidobacterium* and *Scardovia*, and species such as *P. gingivalis, T. forsythia*, and *P. intermedia* have been detected only in dental plaque of toothy centenarians [38], while in edentulous centenarians, the microbiota composition seems to be predominated at the phylum level by *Bacillota* and *Actinomycetota*, while at the genus level, the composition is predominated by *Streptococcus* spp. in both dental plaque and saliva [38].

On the other hand, human gut microbiota in adults is mostly composed of *Bacteroidota* and *Bacillota* (at least 90% of all phylogenetic types) [39], which is further subdivided to consist of more than 100 distinct bacterial species [40]. It is well known that gut microbiota changes with age. In particular, it dynamically changes from birth until three years of age and then becomes more stable [41,42,43,44]. However, during aging, the microbiota composition seems to change (mainly in the diversity of species) once again [45,46,47,48]. In older adults, a decrease in the ratio of *Bacillota*/*Bacteroidota*, a reduced number of bifidobacteria and an increase in certain proteobacteria have been reported [49] (Figure 2).

Interestingly, gut microbiota composition in centenarians has been reported to be more diverse compared to the general older population. In particular, it has been documented that *Bacillota* still represent a significant proportion of the gut microbiota in centenarians. Among *Bacillota*, more beneficial species such as *Lactobacillus*, known for their anti-inflammatory and antioxidant effects, have been reported in Chinese centenarians [52]. In Sardinian centenarians, researchers have reported a depletion of *Faecalibacterium prausnitzii* and *Eubacterium rectale*, but enriched *Methanobrevibacter smithii* and *Bifidobacterium adolescentis* compared with young and older adults [53]. Regarding *Pseudomonadota*, an overall slight increase has been reported with enrichment in *Escherichia coli* [50,53,54]. These changes are not related solely to aging but also to other contributing factors including antibiotic therapy, comorbidities, immune system alterations (e.g., immune senescence), increased intestinal permeability to lipopolysaccharides, and modifications in diet and lifestyle [45,46,55,56,57]. Table 1 compares the main oral and gut microbiota changes across the lifespan.

## 3. Overview on the Interaction Between Oral and Gut Microbiota

The oral–gut microbiota axis is a relatively new field of research, but it is rapidly gaining interest [58]. Although most studies have focused separately on the oral and gut microbiota, emerging evidence highlighted that the two microbiota are interconnected and may influence each other through various mechanisms [59]. The process of oral colonization has been closely linked to the gut microbiota, being anatomically continuous regions in the gastrointestinal tract [59] and through various cross-talk axes, including the oral/lung/gut axis [60,61]. The oral microbiota plays a crucial role in influencing the lung microbiota, primarily due to the translocation of microorganisms through the “bioaerosol” process, which transports them from the oral cavity to the lower airway tract [60,61]. Consequently, the state of eubiosis in the oral microbiota is strongly related to the lung microbiota, which is in turn connected to the gut microbiota and, by extension, to all the microbiota communication axes [60,61].

A recent human study [62] found the presence of 61 amplicon sequence variants (ASVs) in the 96% of participants considered in both the oral and gut microbiota. Of these, 26 ASVs from 18 genera were found in both children and adults, suggesting these microorganisms are persistent colonizers across different life stages. However, this study [62] also emphasized the presence of age-related changes in the microbial composition between children and adults, indicating that microbial diversity remains almost stable until the age of 45, with significant changes thereafter. Notably, the authors reported that 62% of the shared ASVs were more abundant in the oral cavity, indicating that oral-to-gut translocation is a primary mechanism of microbiota transfer between these habitats [62].

Communication pathways between oral and gut microbiota include the enteral, hematogenous and fecal–oral routes. Figure 3 provides an overview of the main mechanisms of interaction between oral and gut microbiota.

### 3.1. The Enteral Route

Each day, nearly 1 to 1.5 L of saliva are produced and swallowed in the human gastrointestinal tract [63] along with ingested food [64]. It has been thus suggested that oral bacteria can be ingested and reach the gut (mainly through saliva), where they may colonize and alter the composition of the gut microbiota. However, since gastric acid and alkaline bile can influence the translocation of oral bacteria, there has been a great debate about whether oral microbiota can colonize the gut. Research has indicated that oral bacteria can migrate to the gut in individuals with weakened oral–gut chemical barriers (such as bile and gastric acid), like infants, people with gastrointestinal disease, or using proton-pump inhibitors (PPIs), and older people [59,64,65]. In fact, it has been reported that the presence of oral bacteria in the gut is more common in older adults compared to younger adults [59,62,66,67], with the use of PPIs resulting in low gastric acidity [65], and with the use of antibiotics [68]. Certain periodontal pathogens, such as *Porphyromonas gingivalis*, *Klebsiella* spp., *Helicobacter pylori*, *Streptococcus* spp., *Veillonella* spp., *Parvimonas micra*, and *Fusobacterium nucleatum*, are capable of surviving in acidic conditions and can thus reach the intestine [11,69]. On the other hand, under normal physiological conditions, some species like *Prevotella* from saliva, have also been found in human stool samples [70]. Additionally, *Helicobacter pylori* infection can disturb the oral microbiota balance and further compromise the gastric environment, promoting the growth of oral bacteria like *Fusobacterium nucleatum* and *Porphyromonas gingivalis* in the human gut [11,71].

Furthermore, some bacteria can produce biofilms or be enveloped by some layers and substances (e.g., mucus) that can potentially give them protection from harsh environments [58]. Biofilm formation enables pathogens like *Streptococcus mutans* to survive in the human oral cavity by utilizing mechanisms such as the production of reutericyclin, which suppresses nearby commensal bacteria and contributes to dysbiosis [11,72].

### 3.2. The Hematogenous Route

Oral bacteria can also spread through the bloodstream to other body sites under certain conditions. Mechanical disruptions like brushing or chewing, inflamed periodontal tissues, or lesions from dental procedures can facilitate the transition of bacteria into the circulatory system through the vascularization and gingival ulceration of periodontal pockets [11,64,73]. Interestingly, some practices such as sleep bruxism (the involuntary grinding or clenching of teeth) can play a role in the migration of oral bacteria in the gut, thus constituting a hypothetical brain–oral–gut axis. In fact, sleep bruxism can lead to oral tissue damage [74], potentially allowing oral bacteria to enter the bloodstream (i.e., oral bacteremia) and then reach the gastrointestinal tract [75]. The presence of oral pathogens can exacerbate inflammation, leading to damage both at oral (i.e., soft and hard periodontal tissues) and at a systemic level [11,64]. As a result, oral bacteria, including *Streptococcus*, *Porphyromonas gingivalis*, and *Fusobacterium nucleatum*, can spread throughout the body, reaching distant organs such as the gut, triggering systemic pro-inflammatory responses and potentially contributing to the pathogenesis of gut diseases [11,64]. In animal models, *Fusobacterium nucleatum*-induced periodontitis has been shown to alter the bacterial microbiota in the gut, promoting intestinal inflammation [76]. In humans, the hematogenous pathway has been documented as the primary route for *Fusobacterium nucleatum* to reach colon tumors, in which it has been associated with chemoresistance and poor prognosis [77]. Virulence factors from periodontal pathogens further enhance inflammation and compromise intestinal epithelial barrier, allowing bacteria and metabolites to leak into the bloodstream, thus strengthening microbial cross-talks throughout the body [11]. The presence of pro-inflammatory oral bacteria, such as *Porphyromonas gingivalis*, in the bloodstream could also affect the gut microbiota, worsening gut permeability, endotoxemia, and leading to metabolic dysregulation and gut dysbiosis, predisposing the body to a vicious cycle of systemic inflammation that damages both microbiomes in both mice models and in humans [63,78,79]. In particular, *Porphyromonas gingivalis* disrupts the colonic epithelial barrier by producing gingipains, which act as mucus-detaching proteases, and by degrading tight junction proteins [64].

*Streptococcus salivarius*, an early colonizer of the oral cavity, can also inhabit the intestinal tract, where it down-regulates the nuclear transcription factor-κB (NF-κB) in small intestinal epithelial cells, contributing to both intestinal inflammation and homeostasis [1].

Augmented levels of some oral bacteria including *Streptococcus mutans* (cariogenic), *Porphyromonas gingivalis* and *Fusobacterium nucleatum* (the main pathogens associated with periodontal disease) have been found in the gut of patients with IBD, Human Immunodeficiency Virus (HIV) infection, liver cirrhosis, and colon cancer [63,64,80,81,82,83]. An elevated presence of *Campylobacter concisus* and *Fusobacterium nucleatum* has also been detected in fecal samples and intestinal biopsies of IBD patients [84] and in human colorectal cancer [85,86]. Furthermore, in IBD, bile acid malabsorption and reduced expression of bile acid receptors have been suggested to facilitate the translocation of oral pathobionts from the oral cavity to the gut [87,88]. Several mechanisms through which oral bacteria can contribute to IBD have been proposed: (1) *Porphyromonas gingivalis* and *Klebsiella pneumoniae* can reduce the expression of tight junction protein-1 and occludin, leading to damage of the intestinal epithelial barrier; (2) *Fusobacterium nucleatum* and *Klebsiella pneumoniae* can stimulate the production of the pro-inflammatory lipopolysaccharide; (3) *Fusobacterium nucleatum* and *Candida albicans* can disrupt T helper (Th)1/Th17 cell balance, leading to inflammatory responses disrupting the host immune system and immune escape induction; (4) *Klebsiella pneumoniae* and *Fusobacterium nucleatum* can migrate to the gut, triggering inflammasome activation in immune cells, which promotes intestinal inflammation [63].

Antibiotics have long been associated with alterations in gut microbiota composition [89]. However, antibiotics are frequently used in dentistry for prophylaxis, to treat infections, and to prevent systemic bacteremia [90]. This, in turn, highlights the importance of communication between the oral and gut microbiota via the bloodstream. Recently, the American Dentistry Association [91] recommended that antibiotic prophylaxis may be indicated before dental procedures only in certain patient subpopulations (e.g., patients with underlying cardiac conditions) and to follow the American Heart Association guidelines for infective endocarditis prophylaxis [92]. Some concerns have been raised regarding antimicrobial resistance, which is accelerating, especially in older adults, being responsible for an increased risk of adverse outcomes and death [93,94]. Inappropriate antibiotic use, as well as poor infection prevention strategies and decreased vaccination rates, have been identified as the main reasons for the global increase in antimicrobial resistance [93]. Indeed, the expert panel of the American Dental Association Council on Scientific Affairs and the Center for Evidence-Based Dentistry suggested antibiotic treatment for target conditions when systemic involvement is present, and prioritization of immediate conservative dental treatment in all cases [95]. In this regard, probiotic supplementation is frequently used as adjunct therapy to prevent antibiotic-induced dysbiosis [96]. Some evidence has formerly documented that certain probiotics can help reducing both occurrence and duration of antibiotic-associated diarrhoea, including the prevention of *Clostridioides difficile*-associated diarrhoea [97]. However, current research does not support the idea that probiotics can fully restore the microbiota to its state before antibiotic use, with limited studies showing that a specific probiotic preparation might delay the recovery of the microbiota after antibiotic disruption [97]. According to a recent systematic review and meta-analysis, probiotic supplementation during antibiotic treatment was not found to be influential on low-diversity dysbiosis [96].

### 3.3. Fecal-Oral Route

Translocation of microorganisms from the gut to the oral cavity can also take place through fecal–oral transmission, either through direct contact or indirectly via contaminated food and beverages. Hands play a crucial role as carriers, facilitating the transfer of fecal and oral microorganisms both within households and between individuals [11,98] as its microbiota overlaps with that of the mouth and the gut [98]. Fecal–oral route transmission is a significant concern, particularly in areas with limited access to clean water, poor sanitation, and hygiene [59]. Immunocompromised individuals and people undergoing radiation therapy for head and neck cancer are also more susceptible to infections through fecal–oral transmission [59]. In particular, radiation therapy has in turn been associated with oral dysbiosis, characterized by increased colonization of opportunistic pathogens such as staphylococci, *Candida* spp., and species from the *Enterobacteriaceae* family, even worsening within poor oral hygiene [99,100].

Additionally, the fecal–oral route facilitates human-to-human pathogen transmission. Enteric viruses, like hepatitis A and E, easily spread via the fecal–oral route in unsanitary conditions and across people, and can significantly disrupt the gut microbiota balance [101,102,103,104]. *Helicobacter pylori*, associated with severe gastrointestinal diseases, can also spread via this route and has been related to hepatitis A infection [105].

### 3.4. Metabolite-Mediated Interactions Between Oral and Gut Microbiota

Recent evidence suggests that metabolites produced by the oral microbiota can influence the gut microbiota and vice versa, although their interplay seems to be quite complex [106]. Key microbial byproducts and derivatives, as well as bioactive components, such as SCFAs, TMAO, indole and its derivates, bile acids, and LPS, appear to play central roles in mediating various pathological conditions, by acting on inflammation, oxidative stress, lipid and glucose metabolism, and body barrier integrity.

SCFAs in the gut may be translocated through the bloodstream to the oral cavity [106]. In turn, the oral microbiota might send signals back to the gut, influencing digestive health. This could indicate that the oral and gut microbiota interact through metabolites to maintain or not the microbial balance. The SCFAs acetate, propionate, and butyrate are produced by gut microbiota starting from undigested dietary fiber [106,107]. The SCFAs that remain unmetabolized in the gut can be circulated to other organs, including the oral cavity in which they can influence pH, through the hepatic portal vein [108]. SCFAs can also be produced by the oral microbiota from carbohydrate hydrolysis or amino acid metabolism [106], although their production in the oral cavity is generally lower compared to the gut [109,110]. These metabolites have anti-inflammatory properties and can influence oral and systemic health in several ways. SCFAs can exert a direct influence on neutrophils by (1) decreasing their production of reactive oxygen species (ROS) and myeloperoxidase and (2) enhancing their apoptosis [106]. In the gut, some amino acids like arginine can be metabolized to compounds like nitric oxide (NO) influencing oral microbiota health through antibacterial function [111]. However, SCFAs seem to play a more complex role in the oral cavity. In fact, although SCFAs are generally produced by oral microbiota as a part of normal local metabolic processes, their overproduction at the oral level has been indicated as a marker of oral dysbiosis, triggering soft tissue damage and an increased inflammatory response in the oral cavity [106,112]. On the contrary, a higher amount of SCFAs in the gut does not damage intestinal epithelium cells, probably because of a better adaptive capacity of the gut mucosa compared to the oral mucosa [106]. At the systemic level, SCFAs have been suggested to exert metabolic effects on obesity and glucose homeostasis [113,114]. SCFAs regulate both metabolism and immune function of the liver mainly by inhibiting histone deacetylases or through the activation of G-protein coupling receptors (GPCRs) [115]. A metagenome-wide association study reported a lower abundance of several butyrate-producing bacteria in fecal samples from patients with type 2 diabetes (T2D) compared with healthy controls, suggesting a potential beneficial effect of butyrate in metabolic diseases [116]. However, there is a paucity of information on its exact role since some studies found a higher amount of butyrate in stool samples of overweight and obese people than in those with a normal weight and a similar diet [117,118,119]. It should also be considered that fecal levels of butyrate might not reflect its physiological concentrations since less than 10% of butyrate produced is excreted with feces [120]. In mice, butyrate seems to improve insulin sensitivity, increase energy expenditure, and promote mitochondrial functioning [121]. Butyrate, by serving as a histone deacetylase inhibitor and ligand to GPCRs, affects cellular signaling in target cells like enteroendocrine cells [122,123]. Indeed, recent evidence indicated butyrate as a new therapeutic target for obesity-related metabolic disorders including T2D [113,122]. Proposed treatment strategies, aimed at increasing its intestinal levels, include supplementation of butyrate-producing bacteria, such as *Faecalibacterium prausnitzii*, and dietary fiber, as well as fecal microbiota transplantation [113,122]. SCFAs also seem to affect lipid metabolism by influencing various pathways in the liver, adipose tissue, and muscle [124,125]. Acetate and butyrate participate in lipogenesis in the liver, where they can stimulate adenosine monophosphate-activated protein kinase (AMPK) phosphorylation and activity, increasing fatty acid oxidation and glycogen storage [124]. On the other hand, propionate most clearly acts as an inhibitor of hepatic lipogenesis by blocking fatty acid synthase expression [124]. Additionally, both acetate and propionate can inhibit intracellular lipolysis, while propionate may also enhance the lipid-buffering capacity in the adipose tissue by increasing the activity of lipoprotein lipase for triglyceride extraction, finally resulting in a reduced lipid overflow and a decreased ectopic fat accumulation, with positive effects on insulin sensitivity [124]. Given their role in regulating inflammation, glucose and lipid metabolism, SCFAs emerged as promising metabolites regulating the pathological process of metabolic dysfunction-associated steatotic liver disease (MASLD) [11,115]. However, the role of SCFAs in regulating lipid metabolism needs further investigation [123,124].

Inflammation, particularly when chronic and of low grade, acts as a key mediator in the impact of oral pathogens on both local and systemic health. Pathogens from the oral cavity can invade the body and elicit immune responses that produce pro-inflammatory cytokines such as tumor necrosis factor alfa (TNF-α), interleukin-1 beta (IL-1β), and IL-6 [126]. These cytokines further exacerbate inflammation in other body sites, leading to vascular damage, endothelial injury, and the formation of atherosclerotic plaques, which can elevate the risk of cardiovascular diseases [126]. In this context, LPSs are important components of the outer membrane of Gram-negative bacteria, found in both the oral and intestinal microbiota, with significant pro-inflammatory effects [115]. LPS are potent activators of the innate immune system. In particular, LPSs bind to Toll-like receptor 4 (TLR4) on immune cells (e.g., macrophages, dendritic cells), activating an intracellular signaling cascade that induces the production of pro-inflammatory cytokines (e.g., TNF-α, IL-1β, IL-6) and chemokines from Kupffer cells [115]. This last mechanism seems also to be triggered by the translocation of *P. gingivalis*, including its LPS, from the oral cavity to the gut, thus leading to increased inflammation and gut dysbiosis [11]. When *P. gingivalis* colonizes the gut, it compromises intestinal barrier function and markedly raises endotoxemia levels by increasing LPS in the bloodstream. The elevation of LPS in the circulation in turn stimulates the expression of flavin-containing monooxygenase 3 (FMO3) and elevates circulating TMAO levels, finally leading to metabolic imbalance, dysbiosis, and inflammation [78,79]. *P. gingivalis* also reduces the expression of tight junction proteins, such as cytosolic zonula occludens-1 (ZO-1) and occludin, in the small intestine, thus enhancing intestinal permeability [78]. Additionally, LPS from *P. gingivalis* stimulate both the NF-κB pathway and Caspase-1 inflammasome, increasing IL-1β and IL-18 production [127], which in turn drive both intestinal inflammation and, by crossing the blood–brain barrier, promote neuroinflammation by activating microglia [128]. *P. gingivalis* also appears to stimulate IL-6 expression via the janus kinase 2/glycogen synthase kinase 3 beta/signal transducer and activator of transcription 3 (JAK2/GSK3-β/STAT3) signaling pathway, which is linked to carcinogenesis and the development of oral squamous cell carcinoma [127].

Furthermore, inflammation may promote mitochondrial dysfunction, leading to the release of mitochondrial damage-associated molecular patterns. This, in turn, triggers increased production of cytokines, chemokines, NO, and ROS, which further exacerbate mitochondrial damage, creating a vicious cycle [129]. This is particularly evident in the older population, in which oral diseases may contribute to chronic systemic inflammation, associated with increased morbidity and mortality [126]. Periodontal pathogens like *Fusobacterium nucleatum* contribute to inflammation by releasing toxic factors, which not only damage the local tissues but also facilitate the progression of diseases like colorectal cancer [126]. Periodontal disease and IBD are both chronic inflammatory diseases, probably interacting through the oral–gut axis. *P. gingivalis*, when it reaches the gut, alters microbiota, promoting inflammation through its interaction with oral-derived Th17 cells transferred to the gut, exacerbating colitis via immune activation and increased levels of IL-1β [130], with subsequent damage to the mucosal barrier [131]. Of note, it has been documented that nearly 10–30% of IBD patients may present oral symptoms both before, during or after gastrointestinal manifestations [132]. This suggests a bidirectional relationship between oral and gut microbiota disruptions in IBD and periodontal disease with consequent inflammatory responses both at the intestinal and at the oral level [132]. Furthermore, oxidative stress in conditions like periodontitis may further trigger inflammatory responses as the levels of salivary oxidants such as malondialdehyde and NO were found to be higher in periodontitis patients [133]. In turn, NO has significant implications for human health, particularly in cardiometabolic diseases such as atherosclerosis as a critical signaling molecule [134]. Being synthesized by endothelial cells, NO may induce relaxation of vascular smooth muscle, therefore leading to the dilatation of blood vessels [134], as well as the inhibition of smooth muscle cell proliferation, platelet aggregation and oxidative stress, with its deficiency being associated with the atherosclerotic process [134]. Oral microbiota, at least to a certain extent, can stimulate the production of NO. In particular, salivary glands absorb nearly 25% of nitrate in food, which is also formed by endogenous oxidation [134]. At the oral level, symbiotic bacteria generate nitrate reductase, metabolizing nitrate (i.e., NO3−) found in saliva into nitrite (i.e., NO2−) [135]. This is the first crucial step in the process of NO conversion in the human body [135]. Subsequently, nitrite can be further reduced to NO [136,137]. Oral genera like *Rothia* and *Neisseria* are known to support vascular function and blood pressure regulation through this nitrate-nitrite–nitric oxide pathway, while genera like *Prevotella* and *Veillonella* seem to impair NO homeostasis [138]. Trimethylamine is produced by the gut microbiota through catabolism of dietary choline, phosphatidylcholine, betaine, and carnitine [139]. Once produced, TMA enters the liver via the portal vein and is oxidized to TMAO by FMO3 [139]. In the human body, TMAO influences multiple metabolic pathways, such as cholesterol metabolism, oxidative stress, immune system regulation, and inflammation, being thus proposed as a novel biomarker of metabolic syndrome [115,134,140]. TMAO can inhibit the synthesis of bile acids, which are crucial for cholesterol elimination, and also disrupt the process of reverse cholesterol transport, leading to the accumulation of cholesterol within macrophages and contributing to the formation of foam cells, a key feature of atherosclerotic plaques [141,142]. As a gut microbiota-derived metabolite, TMAO indirectly promotes atherosclerosis-related inflammation by stimulating ROS and activating signaling pathways involving AMPK and sirtuin 1 [143]. In mice, TMAO also worsens oxidative stress by reducing superoxide dismutase levels, increasing malondialdehyde and glutathione peroxidase levels, and triggering the production of proinflammatory cytokines [144]. Additionally, TMAO amplifies angiotensin II-induced vasoconstriction in mice, further connecting the oral–gut axis to hypertension in both mice and humans [145]. In this context, mice models suggest the oral pathogen *P. gingivalis* can significantly increase levels of endotoxemia through enhancing LPS concentration in the bloodstream, which in turn stimulate FMO3 expression and raise plasma TMAO levels, leading to gut dysbiosis and inflammation, further highlighting the oral–gut communication [79]. For instance, dental treatments to restore oral function in older adults could unintentionally lead to bacteremia, worsening systemic inflammation instead of counteracting it [146]. This highlights the complex role inflammation plays, acting as both a local response to infection and a widespread factor influencing overall health.

Bile acid alterations may also represent a common pathway influencing oral, esophageal and gut microbiota. The identification of several new bile acid receptors and signaling pathways made bile acids a crucial group of metabolites with diverse functions in regulating various aspects of human health, especially for microbiota balance and the mucosal immune system within the intestine [147]. Consequently, disturbances in bile acid metabolism or circulation have been associated with intestinal disorders, including IBD and colon cancer [147]. Physiologically, bile acids favor the digestion of dietary fats by forming micelles, promoting nutrient absorption [148]. Bile acids, mainly through the farnesoid X receptor (FXR), widely expressed in the ileum and the liver, also regulate gene expression in diverse metabolic pathways, including lipid and glucose metabolism, as well as bile acid synthesis itself [115,149,150]. The FXR also mediates anti-inflammatory effects and maintains intestinal barrier integrity. In particular, the FXR, through the down-regulation of the expression of liver X receptor and sterol regulatory element-binding protein 1c seems to reduce fatty acid and triglyceride synthesis in the liver [151]. The modulation of FXR signaling in the gut has thus been suggested as a potential target strategy for both prevention and treatment of the fatty liver and metabolic alterations [152]. Regarding glucose metabolism, it has been reported that mice lacking FXR exhibit both decreased glucose tolerance and insulin sensitivity [149]. Activating FXR with cholic acid seems to reduce glucose levels by suppressing the expression of several liver genes involved in gluconeogenesis [149]. The downregulation of phosphoenolpyruvate carboxykinase expression mediated by the small heterodimer partner highlights the important role of FXR in regulating glucose metabolism [153].

Beyond FXR, Takeda G protein-coupled receptor 5 (TGR5), also known as G protein-coupled bile acid receptor, stimulates glucagon-like peptide-1 (GLP-1) secretion, affecting energy and glucose metabolism, insulin sensitivity and inflammation [154,155]. In particular, the TGR5 modulates inflammatory response via NF-κB signaling pathway and cytokines release (e.g., IL-1β, IL-6, and TNF-α) from macrophages [155,156]. Secondary bile acids at high concentrations have been reported to damage epithelial cells and promote IBD and carcinogenic processes, especially in the colon, through mechanisms like oxidative stress, DNA damage, and resistance to apoptosis [156,157].

Under particular pathophysiological conditions like duodeno-gastro-esophageal reflux, bile acid reflux into the stomach and esophagus can occur, negatively impacting the esophageal epithelium [148]. In gastroesophageal reflux disease (GERD), gastric and duodenal fluids are repeatedly exposed to the esophagus, leading to heartburn and regurgitation [158]. Bile acids, when protonated at acidic pH, have a synergistic damaging effect with gastric acid [159,160]. Saliva may reflect esophageal exposure to bile acids, and their presence is common in conditions like GERD and Barrett’s esophagus [148,161]. Due to their role in Barrett’s esophagus, bile acids have been proposed as diagnostic markers of this condition [148]. Additionally, bile acid alterations influence gut microbiota composition and function, contributing to inflammation and susceptibility to opportunistic infections [162,163]. Elevated bile acid levels may also promote tumor activity in the gastrointestinal epithelium [164]. Bile reflux into the esophagus can damage local mucosa in gastroduodenal disorders [161,164], and high levels of bile acids can activate inflammatory signaling pathways, leading to chronic inflammation in the intestine and colon epithelium [164]. Bile acids have been implicated in the pathogenesis of colorectal and liver cancer, with high-fat Western diets and microbial activity as contributing factors [165]. Primary bile acids are produced by the liver, while gut microbes modify these compounds, enhancing their diversity and biological functions [166]. The microbial metabolism of bile acids regulates both microbial diversity and host physiology, suggesting a bidirectional relationship [166]. At the oral level, recent studies reported an increased total salivary bile acid concentration in GERD patients [148,167]. Krause et al. [167] found higher levels of conjugated salivary bile acids such as glycocholic acid, glycodeoxycholic acid and glycochenodeoxycholic acid in GERD patients, which have been identified as potentially tumorigenic [168]. Other studies also found taurocholic acid in the saliva of GERD patients [148,161], but only glycocholic acid has been associated with dental erosion in this population [161]. An acidic oral environment, as in the case of dental erosion involving the loss of tooth enamel, stimulates protein enzyme production and alters microbial composition, favoring the growth of acid-tolerant bacteria [169,170,171]. This shift can lead to dysbiosis, reducing beneficial bacteria and increasing harmful bacteria linked to caries and periodontitis [169,172,173]. A bidirectional relationship between bile acids and oral microbiota dysbiosis may thus be postulated, as dysbiosis has been associated with GERD [174], Barrett’s esophagus [175], and esophageal cancer [176]. Microbiota alterations in GERD and Barrett’s esophagus, such as an increased abundance of Gram-negative bacteria (e.g., *Fusobacterium*, *Neisseria*, *Campylobacter*, *Bacteroides*, *Proteobacteria*, *Veillonella*) and decreased Gram-positive *Streptococcus*, may thus result from bile and gastric acid reflux [177]. In turn, oral bacteria also play a role in shaping the microbiota of the gallbladder and upper gastrointestinal tract with associations to gallstone disease (GSD) pathogenesis, although the exact mechanisms have not been fully elucidated [178]. *Pseudomonadota*, *Bacillota*, *Bacteroidota*, *Actinomycetota*, *Fusobacteriota*, and *Synergistota* have been found in the bile of GSD patients [179]. Of these, *Pyramidobacter* genus, belonging to the phylum of *Synergistota* and mainly found in the oral cavity, has been found in the bile of GSD patients, further suggesting a role of oral microbiota in gallbladder disease [179]. Poor oral hygiene and missing teeth have been associated with GSD [180], thus suggesting a potential vicious circle among bile acids, dental erosion, oral bacteria and cholelithiasis.

Indole and its derivatives are bacterial metabolites of tryptophan and act as quorum-sensing molecules, regulating bacterial activities like biofilm formation, motility, and virulence [181]. This regulation helps maintain balanced microbial communities in both the oral cavity and gut [181,182]. In fact, the presence of both Gram-positive and Gram-negative bacteria like *Escherichia coli*, *Fusobacterium nucleatum*, *Klebsiella*, *Shigella dysenteriae*, *Vibrio cholerae* and *Enterococcus faecalis*, as well as *Porphyromonas gingivalis*, producing indole in both the oral cavity and the gut, suggests that indole signaling could influence microbial stability and potentially impact overall health [181,182,183]. Indole and its derivatives regulate epithelial barrier integrity, immune responses, and gastrointestinal motility via intestinal receptors [184]. These compounds also enter the liver through the bloodstream, where they influence liver inflammation as well as glucose and lipid metabolism [184]. While indole’s role in the oral cavity is less extensively studied, higher concentrations of indole have been detected in the saliva and gingival crevicular fluid of patients suffering from periodontitis, suggesting indole metabolites produced by oral pathogens might contribute to oral dysbiosis and host inflammation [182]. Indole presence could potentially influence the stability or dispersal of oral biofilms, or modulate the behavior of other oral microbes [181,182,183]. The majority of indole compounds exert their biological effects by activating three primary receptor-mediated signaling pathways: the aryl hydrocarbon receptor (AhR), pregnane X receptor (PXR), and TLR4, which are expressed in various gut cell types, including epithelial cells, fibroblasts, and immune cells [185] and that regulate the expression of genes involved in immunity, inflammation, and intestinal barrier function [115]. Indole plays a protective role in the gut by strengthening epithelial tight junctions and reducing inflammation and tissue injury [115]. It also modulates GLP-1 secretion in colonic L cells, enhancing calcium influx and slowing GLP-1 breakdown [186]. Key indole derivatives include the indole-3-aldehyde (IAld), Indole-3-acetic acid (IAA) and Indole-3-propionic acid (IPA) [115]. The IAld produced from indole pyruvate by aromatic amino acid aminotransferase acts as a ligand for the AhR. It stimulates IL-22 production, protecting against mucosal damage and candidiasis [187]. IAld also promotes IL-10 receptor expression, supporting anti-inflammatory pathways [188]. Levels of IAA are reduced in individuals with metabolic syndrome and obesity, correlating with glucose intolerance and liver steatosis [115,189]. IAA decreases pro-inflammatory cytokines like TNF-α, monocyte chemoattractant protein-1, and IL-1β, reduces free fatty acid synthesis, and inhibits lipogenesis in an AhR-dependent manner, suggesting a protective role against MASLD [190]. As a PXR ligand, IPA suppresses intestinal inflammation and enhances gut barrier function [191]. It also acts as an antioxidant, protecting brain, neuronal, and liver cells from oxidative damage [115]. Recent findings show IPA improves glucose metabolism by lowering blood glucose and insulin levels in rats, indicating new potential targets for insulin resistance [192].

In this context, autoinducer 2 (AI-2) is a universal signaling molecule produced by many bacterial species, including oral bacteria like *A. naeslundii* and *S. oralis*, *S. mutans*, *P. gingivalis*, and *F. nucleatum* [193,194,195], that facilitates interspecies communication and coordinates biofilm formation, virulence, and motility [183,195]. AI-2-promoted coaggregation and biofilm maturation have been suggested as key mechanisms for dental plaque development, contributing to dental caries and periodontal diseases [193]. Indole and AI-2 can regulate overlapping bacterial functions but often have opposite effects. To date, in some bacteria, AI-2 promotes biofilm formation, while indole inhibits it [183]. This interplay helps fine-tune microbial community structure and behavior [183].

The balance between AI-2 and indole could therefore influence the overall structure and pathogenicity of both the oral and gut microbiota, by influencing the transition from commensal to pathogenic states in bacteria, affecting diseases in the oral cavity and gut.

## 4. Dietary Strategies

Diet plays a significant role in the modulation of both oral [196,197] and gut microbiota [198,199]. Fermentable carbohydrates, such as simple sugars and starches, serve as primary energy sources for bacterial metabolism [200]. Indeed, diets rich in carbohydrates, particularly in refined sugars, contribute to increased dental plaque accumulation exacerbating the proliferation of cariogenic bacteria like *S. mutans* and *F. nucleatum* [201,202]. Westernized diets characterized by refined grains and ultra-processed foods and low in fruits and vegetables, and thus poor in micronutrients, predispose the body to a greater pro-inflammatory state both at the periodontal tissue [9] and systemic level [203]. Although the association between carbohydrate intake and periodontal disease is less well studied [204], emerging evidence suggests that dietary patterns emphasizing whole grains over refined carbohydrates may confer periodontal benefits [73,205,206]. Certain nutrients, including dietary fats and vitamin C, appear to support *Fusobacteria*, while dietary fiber and dairy products have been associated with oral microbial homeostasis [6,207,208,209,210]. Dietary fiber is a key component of healthy dietary patterns, including the Mediterranean diet [211]. As previously mentioned, dietary fiber is metabolized by the gut microbiota into SCFAs, which serve as an energy source for colon cells, help maintain gut barrier integrity, and regulate inflammation. This anti-inflammatory effect can positively impact oral health by reducing periodontal inflammation, while also influencing overall metabolism and immune function [206,212]. Diets low in fiber and high in ultra-processed foods can often lead to the alteration of intestinal barrier integrity and increased gut permeability, allowing the translocation of harmful bacterial products, such as oral pathogens (e.g., *F. nucleatum*, *P. gingivalis*) and LPS, to spread through the bloodstream, triggering low-grade systemic inflammation [128,213,214]. Increasing fiber intake has been shown to enhance butyrate-producing gut bacteria and to decrease *Alloprevotella* at the oral level, highlighting the positive effects of healthy dietary patterns on both oral and gut health [215]. Evidence from preclinical studies also suggests that food additives, despite their impact being difficult to identify within dietary questionnaires, can negatively affect gut homeostasis [211]. In particular, the most widely studied are emulsifiers, which have been associated with a decreased bacterial diversity, an increased amount of pro-inflammatory bacteria such as *E. coli*, as well as with altered microbial gene regulation, decreased mucus thickness, increased gut permeability and activated inflammatory pathways [211]. Dietary fiber also promotes a balanced salivary flow, which helps maintain a stable oral pH, as well as microbial transfer and diversity [11,80,216]. Saliva contains antimicrobial proteins [11,216] and helps the natural cleaning processes of the oral cavity [128]. In fact, a decreased salivary flow can predispose the body to increased colonization by acidogenic and pathogenic bacteria (e.g., *Streptococcus* and *Fusobacterium* genera), further exacerbating pro-inflammatory responses with the abnormal release of IL-6, IL-8, IL-17, IL, 23, IL-1β, TNF-α [216]. Plant-based diets, rich in fibers, vitamins, minerals, and bioactive compounds, play a crucial role in promoting oral and gut health by fostering beneficial microbial communities. These diets support the growth of oral Streptococci such as *Streptococcus sanguinis*, *S. gordonii*, and *S. salivarius*, which are among the earliest colonizers of the oral cavity [217]. Epidemiological evidence indicates that consuming fruits, vegetables, and whole grains significantly lowers the risk of colorectal cancer, largely due to their fiber content, antioxidants, and other bioactive elements such as polyphenols that inhibit cancer cell proliferation and induce apoptosis [218,219,220,221,222]. Plant-based dietary components contribute to a balanced gut microbiota by promoting microbial SCFA production, reducing mucosal inflammation, strengthening the epithelial barrier, and modulating immune responses, thereby supporting immune tolerance and overall gut health [223]. In terms of oral health, plant-based diets have been linked to reduced risk of periodontitis compared to omnivorous diets, likely due to their higher fiber content and lower levels of pro-inflammatory saturated fats, alongside increased polyunsaturated fatty acids (PUFAs) [210,224,225]. This suggests that such diets may enhance the oral–gut–brain axis by promoting healthier microbial communities and reducing inflammation. However, it is worth noting that vegan diets might also pose increased risks for dental erosion and caries, possibly due to lower calcium and vitamin B12 intake and reduced saliva pH [226].

Omega-3 PUFAs are increasingly recognized for their role in modulating both oral and gut microbiota, with important effects on inflammation and oxidative stress. Through their derivatives, the so-called specialized pro-resolving mediators (SPMs), omega-3 PUFAs help resolve inflammation and reduce mitochondrial ROS, supporting tissue health in the oral cavity and gut [227,228,229]. SPMs, produced from dietary omega-3 intake, are present in body fluids such as saliva and gingival tissues, where they interact with immune cells to regulate inflammation in periodontal tissues and promote a healthier balance of oral bacteria [228]. Additionally, in vitro studies suggest direct antibacterial effects of omega-3 PUFAs, inhibiting harmful oral microbes including *S. mutans* and *P. gingivalis* [230,231]. In humans, a randomized trial showed that omega-3 fish oil supplementation during periodontal treatment reduced key periodontal pathogens in advanced periodontitis patients [232]. In the gut, omega-3 PUFAs help maintain microbial balance by supporting the production of SCFAs and modulating inflammation, influencing the ratio of dominant bacterial groups (*Bacillota/Bacteroidota*), which is critical for gut health [229,233]. However, an imbalance between omega-3 and omega-6 intake can disrupt this ratio, potentially contributing to obesity and fatty liver disease [233]. Notably, some long-term human studies have found that omega-3 supplementation does not always produce significant improvements in gut inflammation or cardiovascular outcomes, suggesting effects may depend on factors like dose, duration, and individual health status [227,229].

Excessive alcohol intake has been shown to disrupt both the gut and oral microbiota balance, promoting an overgrowth of Gram-positive species like *S. mutans* while suppressing taxa such as *Fusobacteria* at the oral level [207,234].

Beyond beneficial effects on gut microbiota, growing interest surrounds the use of probiotics and prebiotics to support oral microbiota health [197]. Probiotics, defined as live microorganisms that confer health benefits when administered in adequate amounts, have been extensively studied for their role in oral and periodontal health [235]. Prebiotics are selectively fermentable compounds that promote the growth or activity of beneficial microorganisms [236], often working synergistically with probiotics to enhance their efficacy. While the effects of probiotics on periodontal disease have been widely investigated, the role of prebiotics remains underexplored [217,237].

Prebiotics are typically composed of carbohydrate-based compounds such as fructo-oligosaccharides and galacto-oligosaccharides, but can also include non-carbohydrate substances like polyphenols and PUFAs [236]. These compounds selectively stimulate beneficial taxa such as *Lactobacilli* and *Bifidobacteria*, while inhibiting pathogens like *Clostridia* and *E. coli* [217,236]. Both probiotics and prebiotics exhibit anti-inflammatory and immunomodulatory properties [217,235,236].

Probiotic strains such as *Lactobacilli* and *Bifidobacteria* can help maintain microbial eubiosis, modulate immune responses, and produce antimicrobial agents [238,239]. They exert their effects by competing for epithelial adhesion sites, synthesizing bacteriocins, enhancing immune mechanisms such as secretory immunoglobulin A production, and downregulating pro-inflammatory cytokines and matrix metalloproteinases. These actions contribute to the inhibition of pathogenic bacterial growth and modulation of immune responses both locally and systemically [238,239].

Polyphenols, naturally abundant in fruits, vegetables, and whole grains, are bioactive compounds known for their multifaceted health benefits [218]. Although approximately 90–95% of dietary polyphenols evade absorption in the small intestine due to their large molecular size, they still exert significant biological effects, particularly within the oral cavity and gut microbiota [240,241]. One of the most recognized roles of dietary polyphenols is their antioxidant activity: by scavenging free radicals, they help mitigate oxidative stress [240]. Polyphenol metabolism begins in the oral cavity, where despite incomplete knowledge of the exact metabolic pathways, these compounds modulate the host’s inflammatory response [241,242]. Their chemopreventive potential is demonstrated through the modulation of carcinogenesis processes, notably in colorectal cancer prevention [243,244]. This effect is partly attributed to their antimicrobial activity against oral pathogens such as *Fusobacterium nucleatum* and *Porphyromonas gingivalis*, inhibiting bacterial growth, adhesion to oral cells, and virulence-associated enzymatic activities [241,245,246,247].

Polyphenols positively influence gut health mainly through (1) their prebiotic effect by promoting the growth of beneficial gut bacteria and enhancing the production of SCFAs and (2) their antimicrobial effect including the suppression of pathogenic bacteria by disrupting their structural and functional integrity. This includes the inhibition of penicillin-binding proteins, leading to weakened peptidoglycan cross-linking and increased lysine content. Additionally, polyphenols create an acidic microenvironment via proton donation, impairment of proton pumps, and depletion of bacterial H+-ATPase activity [248].

## 5. Conclusions and Future Perspectives

In conclusion, the interaction between the oral and gut microbiota is a complex and evolving area of research that underscores the interdependence of microbial communities across various body sites. Emerging evidence suggests that the oral and gut microbiota influence each other through several mechanisms, including the translocation of microorganisms via saliva, the fecal-oral route and the bloodstream. Disruptions in oral and/or gut microbiota, such as those caused by oral diseases, antibiotics, or gastrointestinal conditions, can lead to dysbiosis, which in turn may contribute to increased systemic inflammation and a range of pathological conditions, including cardiovascular conditions, IBD, and cancer. Some microbiota-produced metabolites and bioactive compounds, including SCFAs, TMAO, indole and its derivates, bile acids, and LPSs appear to play central roles in mediating various pathological conditions, by acting on both microbiota and influencing inflammation, oxidative stress, lipid and glucose metabolism, and body barrier integrity. Interestingly, some nutrients, bioactive compounds and dietary patterns as well as certain probiotic and prebiotic strains seem to have beneficial effects on both the oral and gut microbiota. However, research on the oral–gut microbiota axis is often limited by observational and cross-sectional study designs, small human sample sizes, and the use of animal models that may not accurately reflect human complexity, factors that collectively hinder the generalizability and interpretation of results. As research continues to uncover the nuances of these microbial interactions, understanding the bidirectional relationship between the oral and gut microbiota is crucial for developing strategies to promote overall health and for disease prevention. Examining combined signaling pathways, molecular mechanisms, and microbial metabolites may offer a more comprehensive understanding of how the oral and gut microbiota influence human health.

## Figures and Tables

**Figure 1 nutrients-17-02538-f001:**
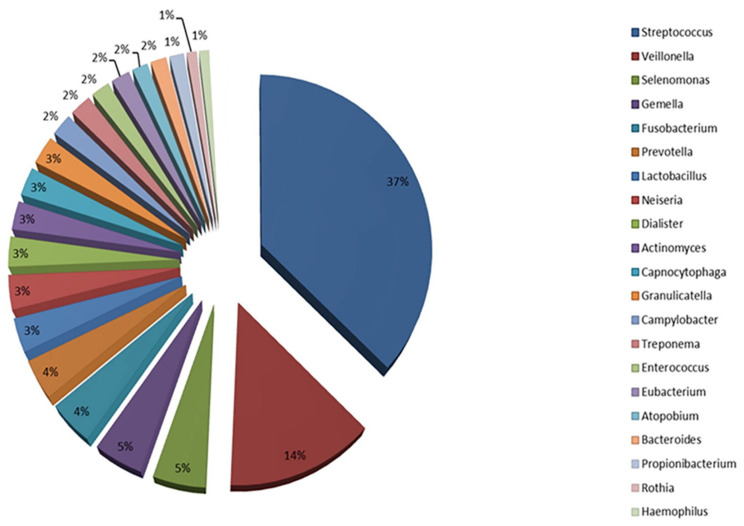
The main relative phyla of oral microbiota. Modified from Santacroce et al. [5] under the terms of the Creative Commons Attribution-NonCommercial 4.0 License (https://creativecommons.org/licenses/by-nc/4.0/, accessed on 5 May 2025), which permits non-commercial use, reproduction and distribution o the work without further permission, provided the original work is attributed. Source: HOMD, http://www.homd.org/ (accessed on 5 May 2025).

**Figure 2 nutrients-17-02538-f002:**
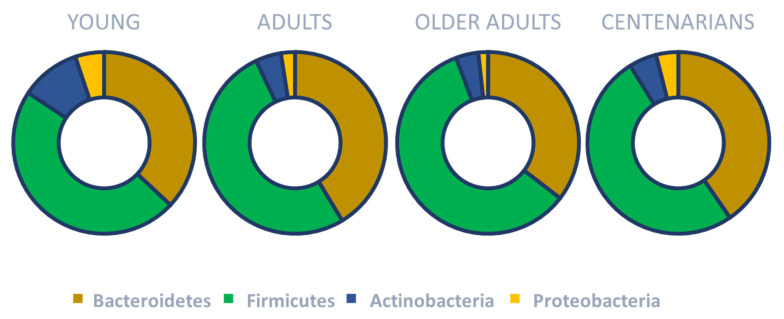
Main gut microbiota changes across the lifespan. Based on concepts and findings of Haran and McCormick [49], Biagi et al. [50] and Monira et al. [51].

**Figure 3 nutrients-17-02538-f003:**
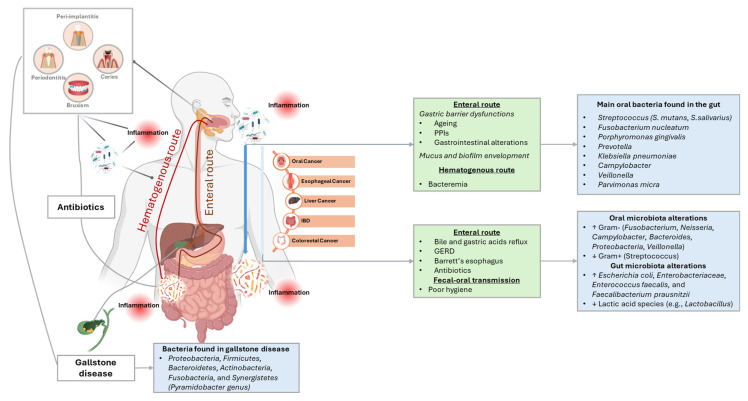
Overview of the three main mechanisms of communication between oral and gut microbiota. (1) Enteral Route: Oral bacteria, such as *Porphyromonas gingivalis* and *Fusobacterium nucleatum*, can reach the gut through saliva, particularly in individuals with weakened oral–gut barriers (e.g., older people, those using PPIs). Bile acids also influence oral, esophageal, and gut microbiota, especially in conditions like gastroesophageal reflux disease and Barrett’s esophagus. Oral bacteria, such as *Pyramidobacter*, also play a role in shaping the microbiota of the gallbladder and upper gastrointestinal tract with associations to gallstone disease pathogenesis. (2) Hematogenous Route: Oral bacteria can enter the bloodstream through disruptions like infections, periodontal disease, dental procedures, or sleep bruxism. This may lead to systemic inflammation and contribute to diseases like IBD and colorectal cancer. Bacteria like *Fusobacterium nucleatum* and *Porphyromonas gingivalis* have been linked to gut inflammation and cancer progression. (3) Fecal–Oral Route: Microorganisms can be transmitted from the gut to the oral cavity through contaminated food, water, or direct contact, especially in areas with poor sanitation or among immunocompromised individuals. This route is associated with the spread of pathogens like hepatitis A, Helicobacter pylori, and enteric viruses, disrupting the gut microbiota. ↑: Abundance; ↓: depletion; IBD: inflammatory bowel disease; PPIs: proton pump inhibitors.

**Table 1 nutrients-17-02538-t001:** Oral and gut microbiota changes across lifespan.

Infancy (0–2 years)		
	Oral Microbiota	Gut Microbiota
**Main composition**	*Streptococcus*, *Staphylococcus*, *Neisseria*	*Enterobacteriaceae*, *Bifidobacterium*, *Lactobacillus*
**Characteristics and influencing factors**	- Aerobes dominate; - Low diversity; - Influenced by delivery mode (i.e., cesarean section vs. vaginal delivery), gestational age (i.e., full-term vs. premature), type of feeding (i.e., breastfeeding vs. formula feeding), maternal nutritional status, exposure to antibiotics.	- Early facultative anaerobes; - Influenced by maternal microbiota, delivery mode (i.e., vaginal delivery vs. cesarean section), type of feeding (i.e., breastfeeding vs. formula feeding), exposure to antibiotics.
**Childhood (2–12 years)**		
	**Oral Microbiota**	**Gut Microbiota**
**Main composition**	*Streptococcus*, *Veillonella*, *Actinomyces*, *Fusobacterium*	Bacillota (e.g., *Clostridium*), Bacteroidota, *Prevotella*
**Characteristics and influencing factors**	- Diversity increases; - Anaerobes appear with tooth eruption; - Influenced by diet (especially high simple sugar intake, solid food), dental problems, infections and oral hygiene.	- Transition to adult-like microbiota; - Influenced by diet (especially high consumption of processed foods, sugar, and fats), gastrointestinal diseases, infections and antibiotics.
**Adolescence (13–18 years)**		
	**Oral Microbiota**	**Gut Microbiota**
**Main composition**	*Streptococcus*, *Fusobacterium*, *Neisseria*, *Prevotella*	Bacillota, Bacteroidota, *Actinobacteria*
**Characteristics and influencing factors**	- Oral microbiota stabilizes; resembles adult profile; - Influenced by hormonal changes, oral hygiene, diet (especially higher in sugars, processed foods, and acidic beverages), use of orthodontic appliances, gingival inflammation (puberty gingivitis), smoking, antibiotic use.	- Bacillota/Bacteroidota ratio continues to shift with growth/puberty; - Influenced by hormonal changes, diet (high consumption of processed foods, sugary snacks and beverages, alcohol), obesity, smoking, infections, use of antibiotics and hormonal contraceptives.
**Adulthood (18–65 years)**		
	**Oral Microbiota**	**Gut Microbiota**
**Main composition**	*Streptococcus*, *Veillonella*, *Actinomyces*, *Prevotella*, *Haemophilus*	*Firmicutes* (e.g., *Clostridia*), *Bacteroidetes*, *Actinobacteria*, *Proteobacteria*
**Characteristics and influencing factors**	- High diversity and relative stability; - Influenced by oral hygiene, sleep bruxism, diet (rich in refined sugars and alcohol), pregnancy, smoking, chronic diseases, medications, xerostomia, infections, gingivitis, antibiotics.	- Stable core microbiome; - Influenced by diet (high in processed foods, sugar, alcohol, and low in fiber), obesity, smoking, infections, chronic diseases, medications, antibiotics.
**Older People (>65 years)**		
	**Oral Microbiota**	**Gut Microbiota**
**Main composition**	**↑** *Lactobacillaceae*, *Streptococcus anginosus*, and *Gemella sanguinis* **↓** *Neisseria*	**↓** Bacillota/Bacteroidota ratio, *Bifidobacteriaceae* **↑** Pseudomonadota (*Escherichia coli*, *Klebsiella*, *Acquabacterium*)
**Main composition**	***Denture users:* ↑** Bacillota and Actinomycetota	-
**Main composition**	***Edentulous:* ↑***Prevotella histicola*, *Veillonella atypica*, *Streptococcus salivarius*, and *Streptococcus parasanguinis*	-
**Characteristics and influencing factors**	- Decreased diversity; - Influenced by denture use, xerostomia, dental problems, oral hygiene, diet (softer, more processed foods with higher sugar and fat content), immune senescence, inflamm-ageing, infections, chronic diseases, polypharmacy, medications, antibiotics.	- Decreased diversity; - Influenced by diet (less fiber and more processed or soft foods due to chewing difficulties or appetite changes), changes in gut motility, reduced digestive secretions, and altered gut barrier function, micronutrient deficiencies, immune senescence, inflamm-ageing, infections, chronic diseases, polypharmacy, antibiotics.
**Centenarians (100+ years)**		
	**Oral Microbiota**	**Gut Microbiota**
**Main composition**	**Toothy centenarians**	**↑** Pseudomonadota (*Escherichia coli* et rel., *Haemophilus* spp., *Klebsiella pneumoniae* et rel., *Leminorella* spp., *Proteus* et rel., *Pseudomonas*, *Serratia* spp., *Vibrio* spp., and *Yersinia* et rel.), Bacillota (*Bacillus* spp., *Staphylococcus* spp.) **↑** *Methanobrevibacter smithii*, *Bifidobacterium adolescentis*, *Clostridium leptum* **↑** Lactic acid species (*Lactobacillaceae*) **↓** Bacillota/Bacteroidota ratio **↓** *Faecalibacterium prausnitzii*, *Agathobacter rectalis*
	***Dental plaque and saliva:* ↑** Spirochaetota and Synergistota (at phylum level), *Aggregatibacter* spp., *Prevotella* spp., *Campylobacter* spp., *Anaeroglobus* spp., *Selenomonas* spp., *Fusobacterium* spp., and *Porphyromonas endodontalis* (at genus level)	
	***Dental plaque:* ↑** *Bifidobacterium* and *Scardovia* (at genus level), *Porphyromonas gingivalis*, *Tannerella forsythia*, and *Prevotella intermedia* (at species level)	
	**Edentulous**	
	***Dental plaque and saliva:* ↑** Bacillota and Actinomycetota (at phylum level), *Streptococcus* spp. (at genus level)	
**Characteristics and influencing factors**	- Toothy centenarians show a more diverse microbiota compared to older adults; - Influenced by oral hygiene throughout life, tooth loss, denture use, diets rich in fiber, low in refined sugars and fermented products, immune senescence, inflamm-ageing, chronic disease, medications, antibiotics.	- More diverse compared to older adults; reduced Bacillota/Bacteroidota ratio compared to older adults - Influenced by genetics, long-term dietary habits, diets rich in fiber, plant-based foods, and fermented products, immune senescence, inflamm-ageing, chronic disease, medications, antibiotics.

↑ Abundance; ↓ depletion.

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
