# Peer review of "The Oral–Gut Microbiota Axis Across the Lifespan: New Insights on a Forgotten Interaction"

_nutrients, 2025, doi:10.3390/nu17152538_

Round 1
Reviewer 1 Report
Comments and Suggestions for Authors
Chronic disorders constitute a significant problem of public health that is associated with an increased risk of mortality and poor quality of life. Malnutrition and poor dietary style have been considered an important problem that worsens the prognosis of patients suffering from metabolic disorders, especially diet-related metabolic dysfunction. The study reported the association of oral-gut microbiome with chronic diseases, which might affect macronutrient (glucose and lipid) metabolism via microbiome regulation in various physical routes. Although the present review reported the possible association of the oral and gut microbiome involved in the route of nutrient digestion and metabolism. However, I do have some questions raised.
- Authors should emphasize the aim of this work in the Introduction section. What is the linkage between oral and gut microbiomes in serum metabolites (i.e., lipid and glucose) and metabolism? Authors should point out and report the possible species and composition of oral and gut microbiome alternation (dysbiosis?) that are involved in macronutrient metabolism but not the gut microbiome species.
- Why do authors only focus on the microbiome but not on their metabolites? Authors should provide updated information on possible microbiome metabolites (i.e., SCFAs, indoles, or secondary bile acids) that might be involved in the oral or gut dysregulation and nutrient-related metabolism.
- Please make sure the connections between the oral and gut microbiomes share the same species of microbiome. The review designed to explain the connection between oral and gut health should be more precise to confirm the potential route and mechanism (oral or gut dysbiosis or barrier protein dysfunction?) However, I did not find such information.
- The connection between oral and gut microbiome data and related parameters was weak and did not point out the connection between diet-related metabolism and microbiome status changes. Data should be reorganized to focus on the interactions between biochemical parameters (glucose and lipid profiles) or related gut microbiome of its metabolites (i.e., SCFA, indole, LPS, and secondary bile acids). Based on this limitation, the topic and conclusion should be reconsidered to be conservative for the reader.
- Overall, this was an important topic, but there was a lack of direct evidence to demonstrate the authors’ aim. The authors should provide a solid connection to convince readers that the oral and gut microbiomes affect diet-related metabolism. Thus, I do not recommend it in its current state.
Author Response
Chronic disorders constitute a significant problem of public health that is associated with an increased risk of mortality and poor quality of life. Malnutrition and poor dietary style have been considered an important problem that worsens the prognosis of patients suffering from metabolic disorders, especially diet-related metabolic dysfunction. The study reported the association of oral-gut microbiome with chronic diseases, which might affect macronutrient (glucose and lipid) metabolism via microbiome regulation in various physical routes. Although the present review reported the possible association of the oral and gut microbiome involved in the route of nutrient digestion and metabolism. However, I do have some questions raised.
Re: Thank you very much for reviewing our manuscript. Your comments and suggestions certainly add value to our paper.
1.Authors should emphasize the aim of this work in the Introduction section. What is the linkage between oral and gut microbiomes in serum metabolites (i.e., lipid and glucose) and metabolism? Authors should point out and report the possible species and composition of oral and gut microbiome alternation (dysbiosis?) that are involved in macronutrient metabolism but not the gut microbiome species.
Re: Thank you. We emphasized the aim of this work both in the introduction and in the abstract as follows: “The aim of this review is therefore to provide an overview of the interactions between oral and gut microbiota, and the influence of diet and related metabolites on this axis.”. We also added some discussion about oral and gut effects (including those mediated by some metabolites produced by both microbiota) on lipid and glucose metabolism.
2.Why do authors only focus on the microbiome but not on their metabolites? Authors should provide updated information on possible microbiome metabolites (i.e., SCFAs, indoles, or secondary bile acids) that might be involved in the oral or gut dysregulation and nutrient-related metabolism.
Re: Thank you. We added a comprehensive discussion about the role of key microbial byproducts and derived bioactive components, such as short-chain fatty acids (SCFAs), trimethylamine N-oxide (TMAO), indole and its derivatives, bile acids, and lipopolysaccharides (LPS) involved in the oral or gut dysregulation and nutrient-related metabolism. We also discussed how these metabolites influence signaling pathways, microbe-host interactions, and key immune mediators like cytokines and transcription factors. We moved the part about bile acids to section 3.4. Metabolite-Mediated Interactions Between Oral and Gut Microbiota.
3.Please make sure the connections between the oral and gut microbiomes share the same species of microbiome. The review designed to explain the connection between oral and gut health should be more precise to confirm the potential route and mechanism (oral or gut dysbiosis or barrier protein dysfunction?) However, I did not find such information.
Re: Thank you. We better clarified this concept by expanding the role of specific metabolites, dietary components and pathogens in the oral and gut microbiota, respectively. We better characterized the connection between oral and gut microbiota also by moving the part about bile acids to section 3.4. Metabolite-Mediated Interactions Between Oral and Gut Microbiota.
4.The connection between oral and gut microbiome data and related parameters was weak and did not point out the connection between diet-related metabolism and microbiome status changes. Data should be reorganized to focus on the interactions between biochemical parameters (glucose and lipid profiles) or related gut microbiome of its metabolites (i.e., SCFA, indole, LPS, and secondary bile acids). Based on this limitation, the topic and conclusion should be reconsidered to be conservative for the reader.
Re: Thank you. As mentioned above, we expanded the discussion about metabolites and dietary compounds in the modulation of both oral and gut microbiota, including their effects on lipid and glucose metabolism. We also reorganized the conclusion by highlighting the role of microbiota-produced metabolites and bioactive compounds in mediating various pathological conditions, by acting on both microbiota and influencing inflammation, oxidative stress, lipid and glucose metabolism, and body barrier integrity. We added some limitations and future perspectives in the conclusion section as follows: “However, research on the oral–gut microbiota axis is often limited by observational and cross-sectional study designs, small human sample sizes, and the use of animal models that may not accurately reflect human complexity, factors that collectively hinder the generalizability and interpretation of results.” and “Examining combined signaling pathways, molecular mechanisms, and microbial metabolites, may offer a more com-prehensive understanding of how the oral and gut microbiota influence human health.”.
5.Overall, this was an important topic, but there was a lack of direct evidence to demonstrate the authors’ aim. The authors should provide a solid connection to convince readers that the oral and gut microbiomes affect diet-related metabolism. Thus, I do not recommend it in its current state.
Re: Thank you for your helpful comments. We hope our revision will now be satisfactory.
Reviewer 2 Report
Comments and Suggestions for Authors
The review appears quite concise and lacks sufficient depth for publication in a high-impact journal. I have a few additional comments that should be addressed. The authors mention a range of diseases influenced by the Oral–Gut Microbiota Axis, but it’s unclear how this axis impacts each condition. Please clarify whether the supporting evidence comes from animal studies or human data, and consider including a table summarizing these studies.
Additionally, it would be helpful to compare the key factors affecting both the oral and gut microbiomes. Including relevant signaling or metabolic pathways involved in the Oral–Gut Axis would also enhance the scientific depth of the manuscript.
Author Response
The review appears quite concise and lacks sufficient depth for publication in a high-impact journal. I have a few additional comments that should be addressed. The authors mention a range of diseases influenced by the Oral–Gut Microbiota Axis, but it’s unclear how this axis impacts each condition. Please clarify whether the supporting evidence comes from animal studies or human data, and consider including a table summarizing these studies.
Re: Thank you for the time spent reviewing our manuscript and for your valuable suggestions to improve our manuscript. We hope that our revision, based on your feedback, has improved the quality of the work. We also added some discussion about oral and gut effects (including those mediated by some metabolites produced by both microbiota) on lipid and glucose metabolism, as well as their influence in various pathological conditions. We added a comprehensive discussion about the role of key microbial byproducts and derived bioactive components, such as short-chain fatty acids (SCFAs), trimethylamine N-oxide (TMAO), indole and its derivatives, bile acids, and lipopolysaccharides (LPS) involved in the oral or gut dysregulation and nutrient-related metabolism. To better characterize the work, we moved the part about bile acids to section 3.4. Metabolite-Mediated Interactions Between Oral and Gut Microbiota. We also emphasized through the paper, the studies based on animal versus human models by adding to each finding if studies were conducted in humans, in mice, etc..
Additionally, it would be helpful to compare the key factors affecting both the oral and gut microbiomes. Including relevant signaling or metabolic pathways involved in the Oral–Gut Axis would also enhance the scientific depth of the manuscript.
Re: We discussed how key microbial products and bioactive compounds influence signaling pathways, microbe-host interactions, and key immune mediators like cytokines and transcription factors influencing pathological conditions.
Reviewer 3 Report
Comments and Suggestions for Authors
Although this review article is appropriately prepared, organized and written, its focus is not how nutrition interacts with oral-gut microbiota axis. The only section that is partly associated with "nutrients" and "nutrition" is section 4.
As a result, I think that this review article, as it is now, is not appropriate for this journal. Please select another journal or extensively reorganize it to make it appropriate for "Nutrients" journal.
Author Response
Although this review article is appropriately prepared, organized and written, its focus is not how nutrition interacts with oral-gut microbiota axis. The only section that is partly associated with "nutrients" and "nutrition" is section 4.
As a result, I think that this review article, as it is now, is not appropriate for this journal. Please select another journal or extensively reorganize it to make it appropriate for "Nutrients" journal.
Re: Thank you for the time spent reviewing our manuscript and for your suggestion to improve our manuscript. We hope that our revision, based on your feedback, has improved the quality of the work. Specifically, we expanded our manuscript by discussing the role of key microbial byproducts and derived bioactive components, such as short-chain fatty acids (SCFAs), trimethylamine N-oxide (TMAO), indole and its derivatives, bile acids, and lipopolysaccharides (LPS) involved in the oral or gut dysregulation and nutrient-related metabolism. We also added some discussion about oral and gut effects (including those mediated by some metabolites produced by both microbiota) on lipid and glucose metabolism, as well as their influence in various pathological conditions. We emphasized the role of the diet in the introduction section: “In this context, diet plays a pivotal role in shaping both the oral and gut microbiota composition. The variety of dental problems associated with oral dysbiosis, alter food choices leading to the consumption of soft and easy-to-chew foods, frequently rich in saturated fats and refined sugars but lacking essential beneficial nutrients (e.g., proteins, fiber, vitamins, and minerals) [8–10] with the consequent disruption of gut microbiota homeostasis [11]. This is particularly evident in the older population, which frequently experiences dental problems associated with oral dysbiosis, dry mouth (e.g., xerostomia), and increased vulnerability to infections negatively affecting gut microbiota health, bolus formation, and contributing to the so-called anorexia of aging [12,13]. Some nutrients, bioactive compounds and dietary patterns as well as certain probiotic and prebiotic strains seem to have beneficial effects on both the oral and gut microbiota.”
We also highlighted the role of dietary fiber, polyunsaturated fatty acids and polyphenols as follows: “Dietary fiber is a key component of healthy dietary patterns, including the Mediterranean diet [212]. As previously mentioned, dietary fiber is metabolized by the gut microbiota into SCFAs, which serve as an energy source for colon cells, help maintain gut barrier integrity, and regulate inflammation. This anti-inflammatory effect can positively impact oral health by reducing periodontal inflammation, while also influencing overall metabolism and immune function [207,213]. Diets low in fiber and high in ultra-processed foods can often lead to the alteration of intestinal barrier integrity and increased gut permeability, allowing the translocation of harmful bacterial products, such as oral pathogens (e.g., F. nucleatum, P. gingivalis) and LPS, to spread through the bloodstream, triggering low-grade systemic inflammation [142,214,215]. Increasing fiber intake has been shown to enhance butyrate-producing gut bacteria and to decrease Alloprevotella at the oral level, highlighting the positive effects of healthy dietary patterns on both oral and gut health [216]. Evidence from preclinical studies also suggests that food additives, despite their impact being difficult to identify within dietary questionnaires, can negatively affect gut homeostasis [212]. In particular, the most widely studied are emulsifiers, which have been associated with a decreased bacterial diversity, an increased amount of pro-inflammatory bacteria such as E. coli, as well as with altered microbial gene regulation, decreased mucus thickness, increased gut permeability and activated inflammatory pathways [212]. Dietary fiber also promotes a balanced salivary flow, which helps maintain a stable oral pH, as well as microbial transfer and diversity [14,97,217]. Saliva contains antimicrobial proteins [14,217] and helps the natural cleaning processes of the oral cavity [142]. In fact, a decreased salivary flow can predispose to an increased colonization by acidogenic and pathogenic bacteria (e.g., Streptococcus and Fusobacterium genera), further exacerbating pro-inflammatory responses with the abnormal release of IL-6, IL-8, IL-17, IL, 23, IL-1β, TNF-α [217]. Plant-based diets, rich in fibers, vitamins, minerals, and bioactive compounds, play a crucial role in promoting oral and gut health by fostering beneficial microbial communities. These diets support the growth of oral Streptococci such as Streptococcus sanguinis, S. gordonii, and S. salivarius, which are among the earliest colonizers of the oral cavity [218]. Epidemiological evidence indicates that consuming fruits, vegetables, and whole grains significantly lowers the risk of colorectal cancer, largely due to their fiber content, antioxidants, and other bioactive elements such as polyphenols that inhibit cancer cell proliferation and induce apoptosis [219–223]. Plant-based dietary components contribute to a balanced gut microbiota by promoting microbial SCFAs production, reducing mucosal inflammation, strengthening the epithelial barrier, and modulating immune responses, thereby supporting immune tolerance and overall gut health [224]. In terms of oral health, plant-based diets have been linked to reduced risk of periodontitis compared to omnivorous diets, likely due to their higher fiber content and lower levels of pro-inflammatory saturated fats, alongside increased polyunsaturated fatty acids (PUFAs) [225–227]. This suggests that such diets may enhance the oral–gut–brain axis by promoting healthier microbial communities and reducing inflammation. However, it is worth noting that vegan diets might also pose increased risks for dental erosion and caries, possibly due to lower calcium and vitamin B12 intake and reduced saliva pH [228].
Omega-3 PUFAs are increasingly recognized for their role in modulating both oral and gut microbiota, with important effects on inflammation and oxidative stress. Through their derivatives, the so-called specialized pro-resolving mediators (SPMs), omega-3 PUFAs help resolve inflammation and reduce mitochondrial ROS, supporting tissue health in the oral cavity and gut [229–231]. SPMs, produced from dietary omega-3 intake, are present in body fluids such as saliva and gingival tissues, where they interact with immune cells to regulate inflammation in periodontal tissues and promote a healthier balance of oral bacteria [230]. Additionally, in vitro studies suggest direct antibacterial effects of omega-3 PUFAs, inhibiting harmful oral microbes including S. mutans and P. gingivalis [232,233]. In humans, a randomized trial showed that omega-3 fish oil supplementation during periodontal treatment reduced key periodontal pathogens in advanced periodontitis patients [234]. In the gut, omega-3 PUFAs help maintain microbial balance by supporting the production of SCFAs and modulating inflammation, influencing the ratio of dominant bacterial groups (Bacillota/Bacteroidota), which is critical for gut health [231,235]. However, an imbalance between omega-3 and omega-6 intake can disrupt this ratio, potentially contributing to obesity and fatty liver disease [235]. Notably, some long-term human studies have found that omega-3 supplementation does not always produce significant improvements in gut inflammation or cardiovascular outcomes, suggesting effects may depend on factors like dose, duration, and individual health status [229,231]. Polyphenols, naturally abundant in fruits, vegetables, and whole grains, are bioactive compounds known for their multifaceted health benefits [219]. Although approximately 90–95% of dietary polyphenols evade absorption in the small intestine due to their large molecular size, they still exert significant biological effects, particularly within the oral cavity and gut microbiota [242,243]. One of the most recognized roles of dietary polyphenols is their antioxidant activity, by scavenging free radicals they help mitigate oxidative stress [242]. Polyphenol metabolism begins in the oral cavity, where, despite incomplete knowledge of the exact metabolic pathways, these compounds modulate the host’s inflammatory response [243,244]. Their chemopreventive potential is demonstrated through the modulation of carcinogenesis processes, notably in colorectal cancer prevention [245,246]. This effect is partly attributed to their antimicrobial activity against oral pathogens such as Fusobacterium nucleatum and Porphyromonas gingivalis, inhibiting bacterial growth, adhesion to oral cells, and virulence-associated enzymatic activities [243,247–249].
Polyphenols positively influence gut health mainly through 1) their prebiotic effect by promoting the growth of beneficial gut bacteria and enhancing the production of SCFAs and 2) their antimicrobial effect including the suppression of pathogenic bacteria by disrupting their structural and functional integrity. This includes the inhibition of penicillin-binding proteins, leading to weakened peptidoglycan cross-linking and increased lysine content. Additionally, polyphenols create an acidic microenvironment via proton donation, impairment of proton pumps, and depletion of bacterial H+-ATPase activity [250].”
Round 2
Reviewer 1 Report
Comments and Suggestions for Authors
The authors answered the considerations smoothly. Good to accept it.
Author Response
Thank you very much for your valuable suggestions and for the time and effort you devoted to reviewing our manuscript.
Reviewer 2 Report
Comments and Suggestions for Authors
The editor may make the final decision on this manuscript. Establishing clear connections between diet and the oral-gut axis is quite challenging. Additionally, the title and the content of the manuscript appear to be somewhat disconnected.
Author Response
Thank you very much for your valuable suggestions and for the time and effort you devoted to reviewing our manuscript.
We agree that the interactions between oral and gut microbiota are quite complex. We have also revised the manuscript according to the editor’s suggestions.